# Feasibility of Optical Genome Mapping from Placental and Umbilical Cord Sampled after Spontaneous or Therapeutic Pregnancy Termination

**DOI:** 10.3390/diagnostics13233576

**Published:** 2023-11-30

**Authors:** Carole Goumy, Zangbéwendé Guy Ouedraogo, Elodie Bellemonte, Eleonore Eymard-Pierre, Gwendoline Soler, Isabelle Perthus, Céline Pebrel-Richard, Laetitia Gouas, Gaëlle Salaun, Lauren Véronèse, Hélène Laurichesse, Claude Darcha, Andrei Tchirkov

**Affiliations:** 1Cytogénétique Médicale, Centre Hospitalier Universitaire de Clermont-Ferrand, CHU Estaing, 63000 Clermont-Ferrand, France; zgouedraogo@chu-clermontferrand.fr (Z.G.O.); e_pierre@chu-clermontferrand.fr (E.E.-P.); g_soler@chu-clermontferrand.fr (G.S.); cpebrel@chu-clermontferrand.fr (C.P.-R.); lgouas@chu-clermontferrand.fr (L.G.); gsalaun@chu-clermontferrand.fr (G.S.); lveronese@chu-clermontferrand.fr (L.V.); atchirkov@chu-clermontferrand.fr (A.T.); 2INSERM U1240 Imagerie Moléculaire et Stratégies Théranostiques, Université Clermont Auvergne, 63000 Clermont Ferrand, France; 3Service de Biochimie et Génétique Moléculaire, CHU Clermont-Ferrand, 63000 Clermont-Ferrand, France; 4CNRS, Inserm, iGReD, Université Clermont Auvergne, 63001 Clermont-Ferrand, France; 5UMR 1095 INRAE/UCA Génétique, Diversité et Ecophysiologie des Céréales (GDEC), Genotyping and Sequencing Plateform Gentyane, 63000 Clermont-Ferrand, France; elodie.bellemonte@inrae.fr; 6Service de Génétique Médicale, CHU Clermont-Ferrand, CHU Estaing, 63000 Clermont-Ferrand, France; iperthus@chu-clermontferrand.fr; 7EA7453 CHELTER Clonal Heterogeneity, Leukemic Environment, Therapy Resistance of Chronic Leukemias, Université Clermont Auvergne, 63000 Clermont-Ferrand, France; 8Service d’Anatomie et de Cytologie Pathologique, Centre Hospitalier Universitaire de Clermont-Ferrand, CHU Estaing, 63000 Clermont-Ferrand, France; helaurichesse@chu-clermontferrand.fr (H.L.); cdarcha@chu-clermontferrand.fr (C.D.)

**Keywords:** optical genome mapping, umbilical cord biopsies, chorionic villi, termination of pregnancy, birth defect

## Abstract

Optical genome mapping (OGM) is an alternative to classical cytogenetic techniques to improve the detection rate of clinically significant genomic abnormalities. The isolation of high-molecular-weight (HMW) DNA is critical for a successful OGM analysis. HMW DNA quality depends on tissue type, sample size, and storage conditions. We assessed the feasibility of OGM analysis of DNA from nine umbilical cord (UC) and six chorionic villus (CV) samples collected after the spontaneous or therapeutic termination of pregnancy. We analyzed quality control metrics provided by the Saphyr system (Bionano Genomics) and assessed the length of extracted DNA molecules using pulsed-field capillary electrophoresis. OMG data were successfully analyzed for all six CV samples. Five of the UC samples did not meet the Saphyr quality criteria, mainly due to poor DNA quality. In this regard, we found that DNA quality assessment with pulsed-field capillary electrophoresis can predict a successful OGM analysis. OGM data were fully concordant with the results of standard cytogenetic methods. Moreover, OGM detected an average of 14 additional structural variants involving OMIM genes per sample. On the basis of our results, we established the optimal conditions for sample storage and preparation required for a successful OGM analysis. We recommend checking DNA quality before analysis with pulsed-field capillary electrophoresis if the storage conditions were not ideal or if the quality of the sample is poor. OGM can therefore be performed on fetal tissue harvested after the termination of pregnancy, which opens up the perspective for improved diagnostic yield.

## 1. Introduction

Genetic diagnosis provides important information for understanding the cause of abnormal development or stillbirth and for counseling on the risk of recurrence. Following intrauterine fetal death (IUFD) or therapeutic abortion, cytogenetic analyses are usually performed on umbilical cord (UC) or chorionic/placental villi (CV) samples. Given the low resolution of karyotypes from UC or CV cultures (~10 Mb), chromosomal microarray (CMA) testing is now recommended in addition to detect smaller chromosomal anomalies in case of abnormal fetal development or unexplained intrauterine fetal death (IUFD) [1,2,3,4]. However, oligonucleotide CMA does not detect triploidies, balanced chromosomal rearrangements (BCRs), or small structural variants (SVs) <50 kb.

OGM is based on the imaging of labelled and linearized high-molecular-weight (HMW) DNA. The images of labelled DNA molecules of each individual genome are converted into maps, which are then assembled into genome scaffolds. A comparison of a patient’s genome assemblies with a reference genome assembly enables the identification of structural variants (SVs) from discordant genome maps.

OGM is able to detect all types of cytogenetic disorders with better resolution than usual cytogenetic techniques and is efficient for clinical diagnosis, in both onco-hematology and constitutional cases of developmental anomalies [5,6,7]. In retrospective studies, OGM showed complete concordance with combined cytogenetic analysis, and identified additional clinically relevant abnormalities [5,8]. OGM is based on the imaging of labelled and linearized high-molecular-weight (HMW) DNA. Its main technical difficulty is obtaining HMW DNA molecules (from 50 Kbp to ≥1 Mbp). The successful isolation of HMW DNA depends on tissue type, quantity, and quality and storage conditions. In this study, we determined the best conditions for isolating HMW DNA from UC and CV sampled after the termination of pregnancy as no protocol exists for this type of biopsy. We also tested pulsed-field capillary electrophoresis to assess the quality of HMW DNA. Finally, we evaluated the performance of OGM using these fetal samples and compared its results to those of routinely used methods.

## 2. Materials and Methods

### 2.1. Sample Collection, Transport, and Storage

Placental/chorionic villi (CV) and umbilical cord (UC) samples were collected after therapeutic abortion, IUFD, or spontaneous abortion in the delivery room or in the pathology laboratory, especially for placental biopsies because the maternal section of the placenta (deciduous) must be differentiated from the fetal section (villi). After sampling, the biopsies were transported to the cytogenetic laboratory in sterile containers with physiological serum or culture medium at room temperature to establish the cell culture required for karyotyping. Only one sample was inadvertently stored for 18 h at 4 °C (#11). The microscopic selection of villi was performed in the cytogenetic laboratory to avoid decidua cell contamination. As soon as the sample arrived in the cytogenetics laboratory or immediately after villous selection, part of it was frozen at −80° for OGM. The time between sampling/collection and freezing is specified for each sample in Table 1. The other part of the samples was used to perform routine cytogenetic diagnosis: CMA (Agilent 60 K, threshold 400 kb, Agilent Technologies, Santa Clara, CA, USA) and/or karyotyping when an aneuploidy or a triploidy was suspected. For CV samples, maternal cell contamination was excluded by short tandem repeats analysis.

### 2.2. Sample Preparation, UHMW DNA Isolation, and Labelling and Chip Loading

UHMW genomic DNA was isolated with the SP Tissue and Tumor DNA isolation kit (Bionano Genomics, San Diego, CA, USA) according to the manufacturer’s instructions (Protocol number 30339). Fetal biopsies were defrosted in stabilizing buffer and then ground with a tissue homogenizer (TissueRuptor^®^). Four samples at most were extracted at any one time.

Before our study we extracted 10 HMW DNA samples from biopsies of different sizes to determine the optimal size needed for each type of tissue to obtain the appropriate DNA concentration (Bionano recommendation: between 35 and 150 ng/µL). After these tests, 15 samples (9 umbilical cord and 6 chorionic villi) of the required size were included in this study (Table 1).

### 2.3. OGM Metrics Data and Data Analysis

Analytical quality control metrics are described in Goumy et al., 2023 [9] and in Bionano Data Collection Guidelines (document 30173). They included the N50 molecule-length, label density (LD), map rate, and effective coverage. The N50 parameter is defined as the optical map contig length that represents 50% of the molecule length distribution. The map rate is the percentage of molecules >150 kbp that map to the reference genome. Effective coverage is the total length of molecules divided by the length of the reference. The throughput target for each sample was 500 gbp.

### 2.4. De Novo Assembly, Structural Variant Calling and Data Analysis

The de novo assembly and variant annotation pipeline were executed with Bionano Solve software v.1.7, as previously described [9].

Data collection typically takes 12 to 14 h. OGM results were analyzed blindly by an independent cytogeneticist and then compared with the results of karyotyping and/or CMA.

### 2.5. Pulsed-Field Capillary Electrophoresis

Because of several mapping failures due to the poor quality of DNA, we tested the Agilent Femto Pulse system, a pulsed-field capillary electrophoresis instrument, to assess the length of extracted DNA molecules. A study conducted by Agilent showed that genomic DNA quality can be accurately assessed to predict the performance of samples with the Saphyr system using the Agilent Femto Pulse system (https://www.agilent.com/cs/library/applications/Copy%20of%20application-gdna-qc-optical-mapping-femto-pulse-5994-0885en.pdf, 2019 accessed on 29 November 2023).

Agilent ProSize data analysis software (5.0.1.3) can quantify and measure the gDNA smear size and calculate the genomic quality number (GQN), a score used to determine the suitability of gDNA samples for use with the Saphyr system. A GQN with a standardized size threshold of 50,000 bp (GQN50 kb) was used, and all fragments <20,000 bp were excluded from the analysis.

We evaluated the quality of the genomic DNA extracted from 17 samples by pulsed-field capillary electrophoresis on a Femto-pulse^®^ instrument (Agilent Technologies, Santa Clara, CA, USA): 6 UC and 6 CV biopsies (corresponding to 12 samples in our study) and 5 control cell cultures for which OGM data had been successfully analyzed. The Femto Pulse was performed retrospectively, and the results were compared to the N50 > 20 kbp parameter of the Saphyr system for the 17 samples (Table 2).

DNA samples were diluted to 0.5 ng/µL in 0.25xTE rinse buffer (Agilent Technologies) and prepared with an Agilent Genomic DNA 165kb kit. Samples were separated on the Femto Pulse system using the Genomic DNA 165kb method with 70 min PFCE.

## 3. Results

### 3.1. High-Molecular-Weight (HMW) DNA Extraction

After testing different biopsy sizes for each type of sample, we obtained an adequate HMW DNA concentration for 15 samples (Table 1).

#### 3.1.1. Umbilical Cord Biopsies

In our laboratory, we mainly use umbilical cord biopsies to perform cytogenetic analyses after pregnancy termination. This procedure is better accepted by the parents than a fetal skin biopsy, especially in case of autopsy refusal. The umbilical cord is poor in cells and essentially composed of a mucoid connective tissue. The extra-cellular matrix protects cells and DNA from degradation. As it is a cell-poor tissue, a larger sample size is necessary to obtain a sufficient amount of DNA. We recommend using biopsies of about 1 cm in length for an umbilical cord of 0.5 cm in diameter, corresponding to approximately 500 mg of tissue. UC samples are difficult to disrupt and to filter because debris clogs the filter. This step is long because of the large size of the biopsy. It is therefore important to keep the sample on ice before and after disruption. We recommend taking a maximum of four samples simultaneously.

We obtained satisfactory DNA concentrations for nine UC biopsies (Table 1). The average N50 > 150 kbp was 260 kp (range: [196–366 kbp]), and the average N50 > 20 kbp was 158 kbp (range: [76–251 kbp]) (Table 1). Saphyr quality control showed a low N50 > 20 kbp and a low LD for four samples (#04, #05, #06, and #07), and a low LD for one (#02) (Table 1). The average map rate was insufficient to realize de novo assembly for these five samples. Poor sample preservation resulted in poor DNA quality and mapping failures, particularly for samples #04 and #06, which were left at room temperature for 120 and 50 h, respectively (Table 1).

#### 3.1.2. Chorionic/Placental Villi Biopsies

Our results showed that CV samples of 10 mg are sufficient to obtain satisfactory amounts of HMW DNA. CV samples are easily disrupted with the TissueRuptor^®^ and the yield of DNA extraction is good even with small samples. DNA concentrations were good for all samples (Table 1).

For CV samples, the average N50 > 150 kbp was 270 kp (range: [207–358 kbp]), and the average N50 > 20 kbp was 176 kbp (range: [86–315 kbp]) (Table 1). Saphyr quality control showed a slightly low N50 ≥ 20 kbp for three samples (#10, #11, and #13), but the LD was right, and the average map rate was sufficient to create a de novo assembly (Table 1).

### 3.2. Evaluation of the Quality of HMW DNA with the Femto Pulse System

We analyzed 17 samples, 12 from the study (6 each of UC and CV), and 5 control cell cultures from CV and amniotic fluid that had been previously successfully mapped. In accordance with Agilent recommendations, we analyzed the GQN50 kb to determine the suitability of gDNA samples for use with the Saphyr system. The GQN50 kb and smear size of each sample are given in Table 2.

Fourteen out of seventeen samples had a GQN50 kb > 6.5. For these 14 samples, the Femto Pulse profiles showed a single narrow peak between 150 and 200 kb, demonstrating high DNA quality (e.g., sample #01, Figure 1A). Two samples could not be analyzed: sample #05, possibly because of a very low LD (10), and sample #07, for which the failure was possibly related to a Flowcell defect.

Three out of 17 samples had a GQN50 kb < 6.5, of which two could not be analyzed (samples #02 and #06). For sample #02, the Femto Pulse profile was atypical with a main peak at 105 kb but also several peaks > 200 kb (Figure 1A). For this sample, labelling was of poor quality (LD = 12.35), which could also explain the failure. For sample #06, the Femto Pulse profile showed that the DNA was highly degraded and therefore of insufficient quality to be analyzed (Figure 1A). Sample #13 could be analyzed despite a GQN50 kb at 3.4. For this sample, the Femto Pulse profile showed that a large proportion of DNA was degraded (peak at 15.726 kb), which explains the low GQN50 kb and the low N50 > 20 kbp, but that there was also a significant number of high-molecular-weight DNA molecules (peak >200 kb), which explains the high N50 > 150 kbp (Figure 1A and Table 2) and probably the success of the analysis.

Figure 1B shows a high correlation between N50 > 20 kbp (Saphyr) and GQN50 kb (Femto Pulse). We excluded sample #05 from this analysis because poor labeling (LD = 10) and not DNA quality was most likely the cause of failure. Twelve samples out of thirteen with a GQN50 > 6.5 could be successfully analyzed (pass). Conversely, two of the three samples with a GQN < 6.5 could not (fail).

### 3.3. OGM Results

Ten de novo assemblies were produced. The time required to obtain results was comparable to that of CMA (4 days), and was much shorter than that for karyotyping (8 to 15 days), which requires cell culture.

Three samples showed abnormalities: three numerical chromosomal aberrations and one CNV. All abnormalities identified on karyotyping and CMA were diagnosed by OGM (Table 3). In seven cases, no abnormality was detected, neither by standard methods nor by OGM. Thus, OGM results were in total agreement with those of standard techniques.

The numerical chromosomal aberrations included one case of trisomy 16, one case of triploidy and one case of monosomy X (Figure 2). The CNV was a 17q12 amplification inherited from the mother, corresponding to a recurrent rearrangement resulting from NAHR and considered as potentially pathogenic with variable expressivity and incomplete penetrance (Figure 2B) [10].

OGM also revealed the presence of numerous small SVs, which was not visible on CMA or karyotyping (Table 3). The median number of SVs per sample was 35 (19–64, interquartile range) with the involvement of 14 (7–27) OMIM genes, of which 4 (2–8) were OMIM morbid genes. We found no SV with already established pathogenic significance.

## 4. Discussion

In this study, we evaluated the feasibility of using OGM for the analysis of UC and CV samples collected after the termination of pregnancy by assessing the quality of the HMW DNA obtained. Only one clinical cytogenetic laboratory has previously evaluated the feasibility of OGM on CV from three products of conception [11]. Our results reinforce those of this initial study and show that OGM can be performed from 10 mg of CV. We could therefore use the technique for prenatal diagnosis based on choriocentesis without the need for prior culture since we receive an average of 40 mg of CV for prenatal diagnosis in our laboratory.

Despite the limited sample size, our study is the first to show a successful OGM analysis of UC samples. Owing to the predominance of extra-cellular matrix and low cellularity, DNA isolation is more difficult for this type of tissue, and we recommend the use of UC biopsies of 500 mg to obtain the required DNA concentration.

All samples kept at room temperature for less than 14 h could be analyzed by OGM (n = 7). Among the samples stored for more than 14 h, only two out of seven could be analyzed. It is therefore advisable to store samples at +4 °C if the time to freezing exceeds 10 h.

IUFD is associated with the maceration of the fetal tissues, which leads to poor DNA quality and limits their use for OGM. In our study, two samples collected following an IUFD were successfully analyzed by OGM. Nevertheless, in such cases, we recommend checking the quality of DNA with the Femto Pulse system before loading on the Flowcell. We show that Femto Pulse GQN50 kb and Saphyr N50 > 20 kbp parameters are comparable. Femto Pulse results can be used to exclude poor-quality DNA samples from OGM analysis and avoid wasting a Flowcell. For the type of samples analyzed in this study, a GQN50 kb >6.5 was successful in over 90% of cases. Conversely, two out of three samples with a GQN50 kb <6.5 could not be analyzed. However, the number of samples analyzed was insufficient to provide an exact GQN50 kb threshold.

In our study, all abnormalities identified by karyotyping and/or CMA were detected by OGM, which confirms that the performance of OGM is comparable to that of the other two techniques combined. This prospective blinded study also showed that OGM does not produce false-positive results, hence its 100% agreement with standard techniques. To our knowledge, this is the first time that a triploidy has been detected by OGM. The detection of triploidy is essential in prenatal care and fetopathology as this anomaly is frequently found in cases of miscarriage and malformation syndrome. An average of 14 additional SVs involving OMIM genes per sample was detected. These SVs are currently underdiagnosed and therefore not referenced in databases. As for CNVs, a database is needed to help in classifying and interpreting the variants. OGM analysis is the next step in the identification of new genes currently not associated with human disease that could have a potential impact on stillbirth or abnormal development. Exome sequencing was recently proposed as a second-line analysis when karyotyping and/or CMA show no abnormality. It can identify sequence variants and some copy number aberrations, but variant calling from short-read sequencing data needs to be improved [12,13]. In addition, SVs involving non-coding regions can have pathogenic consequences by affecting regulatory elements, the organization of topologically associated domains, or splicing mechanisms. OGM can therefore be performed to improve the detection of SV in cases where short-read genome sequencing is negative [14].

## 5. Conclusions

In conclusion, OGM can be performed from UC or placenta biopsies to test for genetic abnormalities after pregnancy termination. Samples should not be stored for more than 14 h at room temperature before freezing. Beyond this time, and for macerated tissue, it is advisable to check DNA quality using the Femto Pulse system. Finally, OGM coupled with exome or genome sequencing has the potential to significantly increase diagnostic yield in genetic analysis following the miscarriage or termination of pregnancy because of developmental abnormalities.

## Figures and Tables

**Figure 1 diagnostics-13-03576-f001:**
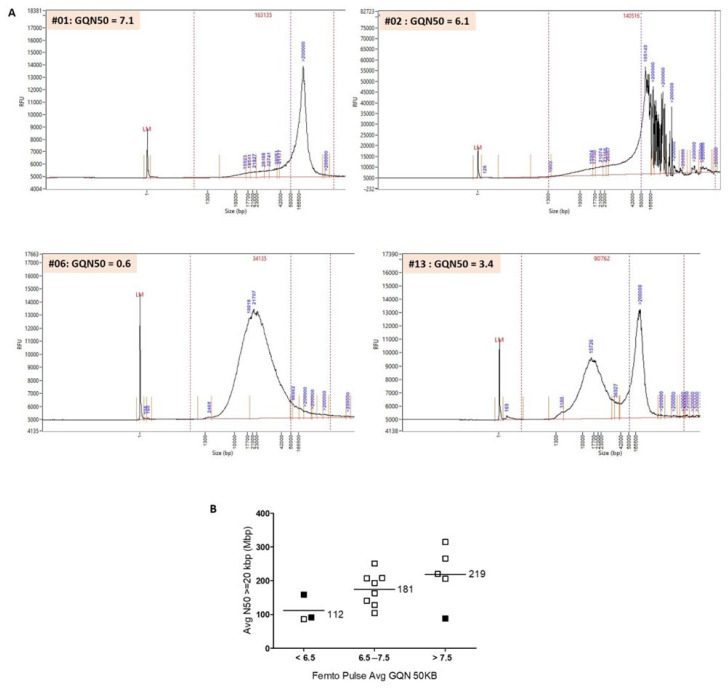
(**A**) Separations of genomic DNA performed on the Agilent Femto Pulse system: examples of profiles obtained. Post-separation analysis was performed with ProSize data analysis software (5.0.1.3). Average smear size was determined by smear analysis (indicated by the dotted lines: range, 1–1.3 to 500–600 kb). Sample #01: average smear size of 163 kb and GQN50 kb of 7.1. A single narrow peak >200 kb shows that DNA is of high quality and can be analyzed by OGM. Sample #02: average smear size of 140 kb and GQN50 kb of 6.1. The profile obtained shows a main peak at 105 kb and several peaks >200 kb. It is difficult to assess DNA quality on this atypical profile. Sample #06: average smear size of 34 kb and GQN50 kb of 0.6. The sample shows extensive degradation and fragmentation and does not meet the necessary criteria for use on Saphyr. Sample #13: average smear size of 90 kb and GQN50 kb of 3.4. We observed two peaks. A large percentage of molecules have a size >150 kb and can be mapped by Saphyr. However, the large amount of degraded DNA can interfere with the analysis. (**B**) Evaluation of the quality of HMW DNA with the Femto Pulse system. Comparison of GQN50 kb of the Femto Pulse and N50 > 20 kbp of the Saphyr. 16 samples are represented in this figure. Sample #05 was excluded from the analysis because very low LD was the cause of the failure (and not the poor quality of the DNA). We classified the samples into three groups, GQN50 kb < 6.5, GQN50 kb between 6.5 and 7.5, and GQN50 kb > 7.5. There was a strong correlation between GQN50 kb and N50 > 20 kb. For samples with GQN50 kb < 6.5 (*n* = 3), the average N50 > 20 kbp was 112 kb. For samples with a GQN50 kb between 6.5 and 7.5 (*n* = 8), the average N50 > 20 kbp was 181 kb. For samples with a GQN50 kb > 7.5 (*n* = 5), the average N50 > 20 kbp was 219 kb. 12 out of 13 samples with a GQN50 kb > 6.5 could be analyzed by OGM (pass = blank boxes). Conversely, of the three samples with a GQN50 kb < 6.5, two could not be analyzed (fail = filled boxes).

**Figure 2 diagnostics-13-03576-f002:**
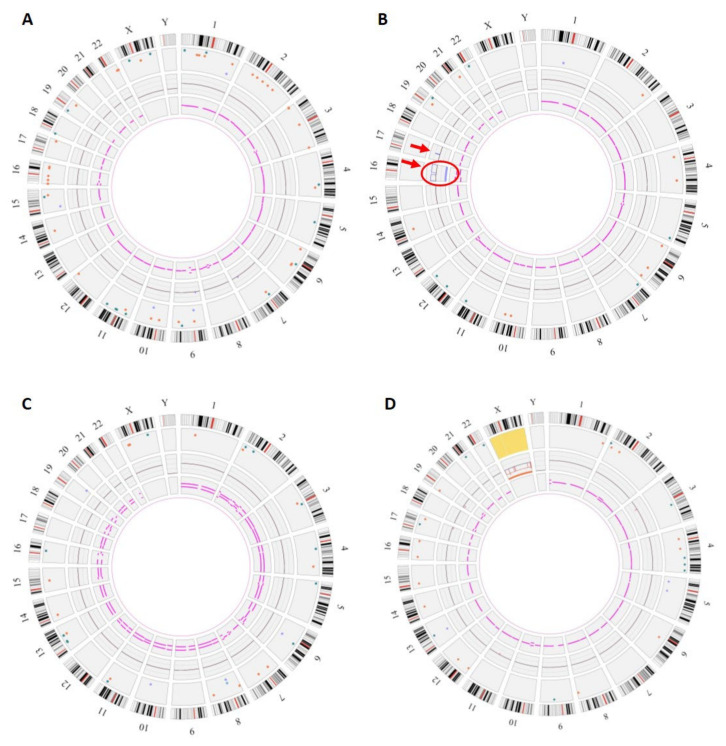
Genomic anomalies identified by OGM. The circos plot is composed of the following layers (from outside to inside): cytoband location, color-coded SV dots, copy-number changes, and variant allele frequency (VAF). In the center of the plot, translocations are shown as lines connecting the genomic loci involved. (**A**) Fetus #1: OGM did not show any copy-number changes or chromosome number abnormalities but detected many SVs (64) (**B**) Fetus #12: Circos plot showed a trisomy 16 (surrounded, fractional copy number: 2.929) and a gain on chromosome 17 at the copy-number changes layer (arrow, VAF: 0.474, Confidence score of 1). (**C**) Fetus #13: Circos plot showed a VAF split for all chromosomes corresponding to a triploidy (VAF~0.33 and 0.66). (**D**) Fetus #14: Circos plot showed a monosomy X (fractional copy number: 1.029).

**Table 1 diagnostics-13-03576-t001:** Type of sample, optimal size/weight of biopsy to obtain satisfactory DNA concentration, type of pregnancy outcome, sample storage conditions, HMW DNA concentration with coefficient of variation and quality control metrics provided by Saphyr system.

Samples Characteristics							Saphyr Quality Control Metrics and Results					
ID	Type	Optimal Size/Weight	Pregnancy Outcome	Storage Time and Temperature before Freezing	HMW DNA Concentration Mean	Coefficient of Variation %	Avg N50 ≥ 150 kbp (Mbp)	Avg N50 ≥ 20 kbp (Mbp)	Avg Label Density (per 100 kbp)	Avg Map Rate %	Effective Coverage	Result
#01	UC	~500 mg L = 1 cm ø = 5 mm	TA	18 h; RT	114.9	44	339	251	18.62	68.2	134X	Pass
#02			TA	20 h; RT	69.8	7	239	159	12.35	53.6	90X	Fail
#03			TA	13 h; RT	36.4	29	366	213	17.75	68.2	115X	Pass
#04			IUFD	120 h; RT	91.3	52	247	76	12.85	9.6	6X	Fail
#05			TA	20 h; RT	78.8	15	198	100	10	17.9	36X	Fail
#06			TA	50 h; RT	61.2	18	210	91	7.54	0.4	0X	Fail
#07			TA	16 h; RT	50.6	23	196	88	13.64	41.4	23X	Fail
#08			IUFD	4 h; RT	58.2	4	278	243	15.04	93.5	156X	Pass
#09			SAB	13 h; RT	80.8	3	271	206	14.46	90.6	149X	Pass
#10	CV	10–20 mg	TA	2 h 30; RT	70.1	8	207	104	14.77	76.6	130X	Pass
#11			TA	72 h; 4 °C	172	5	243	128	15.08	89.4	147X	Pass
#12			IUFD	1 h; RT	145	6	289	220	13.96	86.7	170X	Pass
#13			TA	22 h; RT	89.7	25	227	86	14.6	71.1	139X	Pass
#14			TA	4 h; RT	59	3	299	207	12.47	74.6	121X	Pass
#15			TA	1 h; RT	115.3	9	358	315	14.35	89	150X	Pass

CV: Chorionic villi; L: length; NA: not achieved; IUFD: intrauterine fetal death; RT: room temperature; SAB: spontaneous abortion; TA: therapeutic abortion (fetal anomaly); UC: umbilical cord; ø: diameter.

**Table 2 diagnostics-13-03576-t002:** Saphyr molecule quality control metrics, GQN50 kb, and smear size provided by the Femto Pulse system.

Sample Description		Saphyr Results				Femto Pulse Results	
ID	Type	Avg N50 ≥ 150 kbp (Mbp)	Avg N50 ≥ 20 kbp (Mbp)	Avg Label Density (per 100 kbp)	Issue	Avg GQN_50 kb_	Smear Size (kb)
#01	UC	339	251	18.62	Pass	7.1	163
#02	UC	239	159	12.35	Fail	6.1	140
#05	UC	198	100	10	Fail	8.2	201
#06	UC	210	91	7.54	Fail	0.6	34
#07	UC	196	88	13.64	Fail	8.3	162
#09	UC	271	206	14.46	Pass	7.9	147
#10	CV	207	104	14.77	Pass	7.1	146
#11	CV	243	128	15.08	Pass	6.5	182
#12	CV	289	220	13.96	Pass	7.9	175
#13	CV	227	86	14.6	Pass	3.4	91
#14	CV	299	207	12.47	Pass	6.9	156
#15	CV	358	315	14.35	Pass	8.7	165
	Cell culture control	301	208	13.32	Pass	7.2	162
	Cell culture control	240	141	14.78	Pass	7.5	157
	Cell culture control	228	163	15.4	Pass	7.2	168
	Cell culture control	255	193	13.21	Pass	7.5	223
	Cell culture control	305	266	15.27	Pass	7.9	180

CV: Chorionic villi; UC: umbilical cord.

**Table 3 diagnostics-13-03576-t003:** Fetal phenotype, gestational age, chromosomal abnormalities detected in each sample by OGM, CMA, and karyotyping, and number of structural variants (SVs) detected by OGM with details of number of OMIM and OMIM morbid genes.

Sample Description			OGM Results					Results of Standard Cytogenetic Techniques	
ID	Fetal Phenotype	Gestational Age (WG)	OMG Aneuploidy and Polyploidy	OGM CNVs	Number of SV	Number of OMIM Genes	Number of OMIM Morbid Genes	Karyotype	CMA
#12	IUFD, severe intrauterine growth retardation	22 + 4	Trisomy 16	Gain chr17:34764405-36276597	19	7	2	47,XX,+16	arr[GRCh37] (16)x3,17q12(34817422_36168104)x3 mat
#13	Polymalformative syndrome	17 + 4	Triploidy	No	34	11	2	69,XXX	ND
#14	Cystic hygroma	13 + 5	Monosomy X	No	29	15	6	45,X	ND
#01	Clastic cerebral vascular lesion with voluminous parenchymal destruction, low hair implantation	30	No	No	64	27	8	ND	arr(X,Y)x1,(1-22)x2
#03	Aortic atresia with major hypoplasia of the left heart, hypertelorism, sacral dimple, short corpus callosum	24 + 6	No	No	29	8	3	ND	arr(X,1-22)x2
#08	IUFD, cleft lip	25 + 5	No	No	34	11	3	ND	arr(X,1-22)x2
#09	SAB (recurrence), cystic hygroma, facial dysmorphism	18 + 5	No	No	53	23	7	ND	arr(X,1-22)x2
#10	Facial meningocele	14 + 4	No	No	33	17	5	ND	arr(X,1-22)x2
#11	Acrania, history of spina bifida	13 + 5	No	No	20	7	2	ND	arr(X,1-22)x2
#15	Cardiac chamber imbalance, omphalocele, holoprosencephaly, radial agenesis	34 + 5	No	No	36	18	4	ND	arr(X,Y)x1,(1-22)x2

IUFD: Intrauterine fetal death; SAB: spontaneous abortion; WG: weeks of gestation.

## Data Availability

All data generated or analyzed in this study are included in the published article.

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
