# Peer review of "Feasibility of Optical Genome Mapping from Placental and Umbilical Cord Sampled after Spontaneous or Therapeutic Pregnancy Termination"

_diagnostics, 2023, doi:10.3390/diagnostics13233576_

Round 1
Reviewer 1 Report
Comments and Suggestions for Authors
Manuscript: diagnostics-2711462-peer-review-v1
Title: Feasibility of optical genome mapping from placental and umbilical cord sampled after spontaneous or therapeutic pregnancy termination
The authors of this research presented a study to establish the conditions for using optical genome mapping (OGM) to diagnose chromosomal abnormalities in human placental and umbilical cord samples. I enjoyed this work. I find it very interesting because the authors developed a process to scan the chromosomal maps in depth for these complicated samples. Also, they demonstrated that OMG is an excellent tool to get results about the constitutions of the chromosomes that the conventional tools of karyotypes and chromosomal microarray. I have minimal suggestions to improve the quality of the manuscript:
Lines 84: Why only one sample was kept at 4 °C?
Lines 111-112: Although the authors refer to Goumy et al. 2023, they should describe briefly the OGM metrics.
Lines 153-154: Why did the authors emphasize that sample #01 was kept at room temperature for 18 hours before freezing? I see in Table 1 that there are many differences in the time of hours at room temperature before freezing; these are confusing.
Lines 235-244: The authors should add to Figure 2 a normal OGM result.
Author Response
We thank the reviewer for his favorable review of our manuscript.
Please find below point-by-point responses to the reviewer's comments.
Lines 84: The sample was stored at +4°C by mistake. We have specified this in the manuscript.
Lines 111-112: We have briefly described the OGM metrics: “They included the N50 molecule-length, label density (LD), map rate, and effective coverage. The N50 parameter is defined as the optical map contig length that represents 50% of the molecule length distribution. Map rate is the percentage of molecules > 150 kbp that map to the reference genome. Effective coverage is the total length of molecules divided by the length of the reference. The throughput target for each sample was 500 gbp.”
Lines 153-154: The time interval of room temperature storage prior to freezing is highly variable, depending on the time of fetal expulsion (day, night, weekend) and the time it takes to transfer the samples (longer if the hospital is situated away from our laboratory). We wanted to show that it is possible to analyze a fetal biopsy by OGM even after it has been stored at room temperature for 18 hours. However, we recommend not exceeding 14 hours and testing DNA quality with FemtoPulse prior to processing. To avoid confusion, we removed this sentence.
Lines 235-244: We added to figure 2 a normal OGM result (Figure 2A).
Reviewer 2 Report
Comments and Suggestions for Authors
In this work the Authors aimed to examine the feasibility of using OGM for the examination of UC and CV samples collected after termination of pregnancy by measuring the quality of the HMWDNA gained. The results of this study allow to establish the optimal conditions for sample storage and preparation required for a successful OGM analysis.
The work sounds scientifically original. The design is appropriate. The discussion of the results is convincing. In my opinion the manuscript fit the standard of the journal.
Author Response
We thank the reviewer for his favorable review of our manuscript.